# SceneFunctioner: Tailoring Large Language Model for Function-Oriented Interactive Scene Synthesis

## Abstract

With the Large Language Model (LLM) skyrocketing in recent years, an increasing body of research has focused on leveraging these models for 3D scene synthesis. However, most existing works do not emphasize homeowner's functional preferences, often resulting in scenes that are logically arranged but fall short of serving practical functions. To address this gap, we introduce SceneFunctioner, an interactive scene synthesis framework that tailors the LLM to prioritize functional requirements. The framework is interactive, enabling users to select functions and room shapes. SceneFunctioner first distributes these selected functions into separate areas called zones and determines the furniture for each zone. It then organizes the furniture into groups before arranging them within their respective zones to complete the scene design. Quantitative analyses and user studies showcase our framework's state-of-the-art performance in terms of both design quality and functional consistency with the user input.

## 1 Introduction

Synthesizing 3D indoor scenes has become a widely explored topic over the past decade (Zhang et al., 2019a). A substantial research body focuses on automatically generating appropriate furniture and layouts using various approaches such as optimization (Weiss et al., 2018), relation priors and scene graphs (Zhang et al., 2021b; Gao et al., 2023), and learning-based frameworks (Paschalidou et al., 2021; Tang et al., 2024; Sun et al., 2024). Concurrently, there is growing interest in user-controlled scene synthesis that tailors the generation to user preferences and more practical scenarios. A notable research branch addresses the interactive synthesis of indoor scenes (Yu et al., 2015; Zhang et al., 2019b; 2023). These studies often incorporate user input directly through an interface (e.g., a control panel) and integrate this input into the generation process. Recently, more methods have emerged that enable natural inputs, such as text, to control the generation (Yang et al., 2021; Hwang et al., 2023). Leveraging the exceptional comprehension and generation capabilities of Large Language Models (LLMs) (Wei et al., 2022; Zhao et al., 2023), many LLM-assisted scene synthesis frameworks have been developed (Fu et al., 2024; Lin & Mu, 2024; Çelen et al., 2024). Integrating the LLM allows the user input to be seamlessly converted to design schemes.

Nowadays, economic realities have fueled a trend toward residential rooms that serve multiple functions (e.g., living, storage, and relaxing), stressing the importance of function-oriented designs (Kim et al., 2011; Zandieh et al., 2011; Dai & Mu, 2023). Homeowners expect rooms to serve specific practical functions rather than merely featuring a reasonable layout and furniture arrangement. For example, typical interactive frameworks suggested the furniture based on preprocessed priors such as spatial relations (Zhang et al., 2021a) and neural representations (Zhang et al., 2019b), while the functional needs, i.e., functional priors, are yet to be concerned. As a result, users may struggle to achieve their desired functions, even though the layouts are plausible and aesthetic. In LLM-based text-controlled approaches, users can express their functional requirements through text input. However, without sufficient reference information on these functions and function-oriented prompting, it remains challenging for the LLM to implement these functions in the final design. For example, without related information, the LLM may not know what an "art design" function means. Moreover, simultaneously handling functional requirements and other given goals/constraints (e.g., decid-

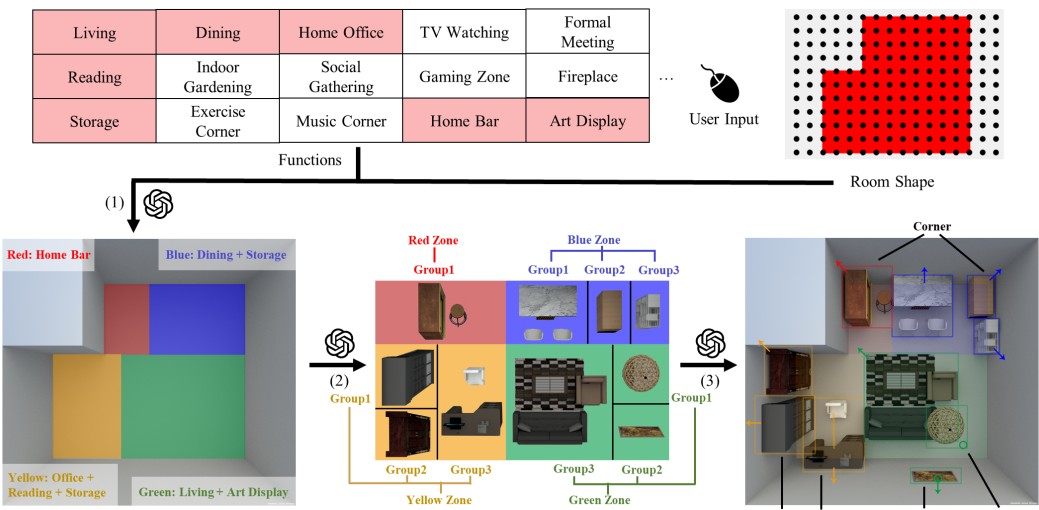

Figure 1: We propose an interactive framework for synthesizing scenes based on user-specified residential functions. The user selects functions from the candidates and sketches the room architecture as the input (Top). The framework then follows a three-step process: (1) The functions are distributed across multiple "zones" (Left-Bottom). In this example, the red zone stands alone for a home bar, while the green zone serves both living (featuring a coffee table, sofa, carpet, etc.) and art display (featuring a painting) functions. (2) Within each zone, furniture items are arranged locally as groups (Middle-Bottom), such as a dining table with two chairs. (3) Furniture groups are arranged relative to their zones (Right-Bottom), e.g., placed against the border or in a corner of a zone. Please refer to our supplementary video for a quick overview of our method and interactive demos.

ing style, local layout, and furniture categories) can potentially reduce the LLM's performance (Liu et al., 2024).

This paper proposes SceneFunctioner, a function-oriented scene synthesis framework incorporating user interaction. We focuses on individual-room-leveled scenes. As shown in Figure 1, users can select their room functions in order of priority and customize the room's shape. Afterwards, our framework employs the LLM to process these inputs and generates a scene that adheres to the user-specified functions. In order to address the challenges described above, we follow three ideas to tailor the LLM in implementing our framework:

First, rather than having the LLM manage all functions in a single step, we introduce "zones" to divide and organize these functions. A zone represents an area containing furniture serving one or more specific functions, ensuring consistency of the functions within it. With the room divided into separate zones, each fulfilling particular functions, the LLM can easily comprehend and execute the synthesis task while following the functional requirements.

Second, we break down the generation task into three sequential steps to further reduce the complexity for the LLM. The first step determines the zones and assigns their respective functions. Then, the subsequent steps focus on furniture layout within each zone, with the second step arranging furniture locally as groups and the third step placing these groups within the zone.

Finally, we design postprocessing and feedback mechanisms to address potential errors the LLM makes, such as incorrect formatting, object collisions, or logical inconsistencies. As illustrated in Figure 2, only when the postprocessed checks pass can the framework proceed to the next step. These mechanisms allow the LLM to improve its response iteratively and further enhance the scenes.

We conduct quantitative analyses to evaluate SceneFunctioner. Compared with LayoutGPT (Feng et al., 2024) and I-Design (Çelen et al., 2024), our method excels in generating state-of-the-art scenes that meet functional needs and ensure practicality. Additionally, a user study involving interactive design with SceneFunctioner demonstrates its effectiveness in producing satisfactory scene quality while significantly reducing design time.

Our work features the following contributions:

- We present an interactive scene synthesis framework that prioritizes the user's preferences for room functions.
- We structure the task into three steps and implement verification and feedback mechanisms at each step, contributing to manageable and reliable LLM-based scene synthesis.
- We propose using zones as units that decompose the functions in a room and serve as a bridge for the LLM to organize the functions and furniture arrangement effectively.

## 2 RELATED WORKS

### 2.1 INTERACTIVE SCENE SYNTHESIS

Interactive scene synthesis generally involves suggesting or editing furniture in a scene based on user inputs. Yu et al. (2015) introduced the Clutterpalette, which suggests small-scaled items when the user points to a location in the scene, enhancing scene details. Similarly, Zhang et al. (2021a) developed a framework that enables real-time inference of furniture based on cursor movements and clicks. They (Zhang et al., 2023) further expanded it to support editing multiple objects simultaneously. Yan et al. (2017) presented an intelligent editing system that automatically refines the layout whenever the user moves the furniture. Ma et al. (2018) leveraged semantic scene graphs for language-driven scene generation and editing. Zhang et al. (2019b) utilized an interface that asks for user preferences, such as furniture category and relations, to customize small objects more effectively. Recently, Zhang et al. (2024) proposed a novel system that allows the user to edit the floor plan while suggesting furniture arrangements in real time. These methods depended on preprocessed priors and did not necessarily generate scenes that satisfy functional requirements. There were other interactive works for generating scenes, such as converting the user's sketch into a well-arranged 3D scene (Xu et al., 2013) and diffusion-based 3D content generation through a 3D creator interface (Li et al., 2024b). However, they did not align with our task of selecting and arranging furniture.

### 2.2 LLM-ASSISTED SCENE SYNTHESIS

The LLM can enhance the scene synthesis task by providing direct (e.g., positions, sizes, and styles) and indirect (e.g., scene graphs and spatial relations) guidance for furniture, layouts, and floor plans. Feng et al. (2024) selected example layouts from a database to instruct the LLM in generating layouts with specified furniture sizes and positions. Yang et al. (2024a) improved this LLM-assisted method by incorporating spatial relations and enabling user editing. Çelen et al. (2024) proposed an LLM-assisted interior design pipeline that supports communication between the LLM and the user, as well as among multiple LLM agents, for iterative layout refinement. Yang et al. (2024b) built a system for generating house-scale indoor environments, tailoring the LLM to determine the floor plan, doorways, furniture, and overall layout.

Entrusting the LLM with a complex generating task can introduce challenges and give rise to errors (Liu et al., 2024). In order to mitigate it, some studies implemented refinement approaches to improve the LLM's response. For example, Aguina-Kang et al. (2024) employed force-based layout optimization and error correction on the plan produced by the LLM. Zhou et al. (2024) applied a global scene optimization process. Fu et al. (2024) facilitated diffusion models to correct the object placement and perform texture inpainting to improve the results.

However, as introduced in Section 1, the above approaches are not tailored to address user preferences regarding the functional aspects of a scene. This paper tackles this issue by explicitly allowing users to select desired functions and precisely instructing the LLM to adhere to these functions.

## 3 METHOD

### 3.1 OVERVIEW

Our framework yields an indoor scene faithful to the user-specified functions and room shape, as illustrated in Figure 2. It first addresses zones (Section 3.2), where the LLM decides their corre-

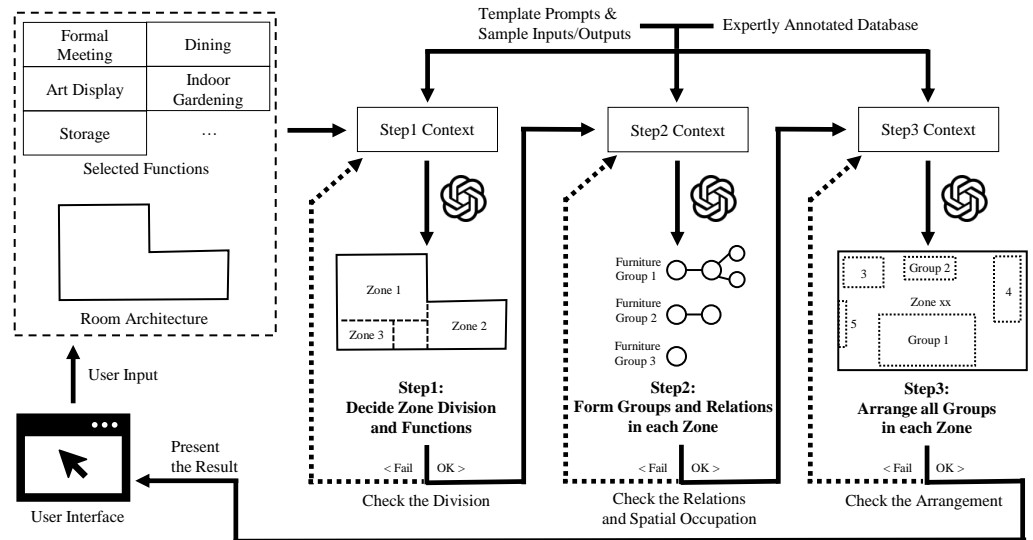

Figure 2: The overview of our interactive framework. Based on given functions and a room shape, our framework distributes the functions into several zones (Step 1). Next, it divides all furniture objects within each zone into several groups and establishes graph-based relations within each group (Step 2). Finally, it arranges these furniture groups within each zone into an appropriate layout to complete the final scene (Step 3).

sponding shapes, functions, and furniture objects. We then check if the room can be appropriately divided into these zones. The second step (Section 3.3) groups and arranges furniture locally. The LLM is tasked with dividing each zone's furniture objects into several groups while using a graph structure to describe the object relations in each group. We then check if these groups are logically valid and spatially collision-free. Finally, the last step (Section 3.4) arranges these furniture groups within each zone, anchoring them to zone borders, corners, ceilings, etc.

## 3.2 DECIDING ZONES AND FUNCTIONS

The allocation and implementation of functions are critical in multi-functional design (Dai & Mu, 2023). Typically, different functions are treated separately to prevent unnecessary interference. For example, a designated area may be exclusively reserved for dining. However, certain cases allow multiple closely related functions to be treated together, particularly when they share the same furniture objects. For instance, an area with two sofas and a coffee table can function both as a relaxation space and for formal meetings. Given the complexity of accommodating various functions within an irregular room, it may be difficult for the LLM to effectively handle the entire task in a single shot. Therefore, we structure the task into two layers to reduce the complexity: zones within the room (this step) and furniture within the zones (the latter two steps). We define a "zone" as an independent area designated for one or multiple closely related functions. While zones may be spatially adjacent, their functions are not necessarily interconnected.

This step addresses the zones and their attributes. First, a **context** is established using the user input, predefined prompts, sample inputs/outputs, and annotated data. The LLM is then queried to generate a **response**. This response is subsequently parsed and **postprocessed**. Our framework attempts to place the zones in the room. If the placement succeeds, our framework proceeds to the next step; if not, **feedback** is provided to the LLM for a revised response. To be more specific:

**Context.** The LLM is provided with the following information: (1) the selected functions ordered by priority, (2) the room shape represented by a point list, (3) annotated descriptions of the selected functions (e.g., "Formal Meeting emphasizes formal hosting for guests, usually incorporating business-oriented furniture with a professional layout"), and (4) annotated data suggesting appropriate furniture for these functions. The LLM is instructed to thoughtfully balance the distribution of functions into zones and consider implementable solutions for the room. Additionally, we append

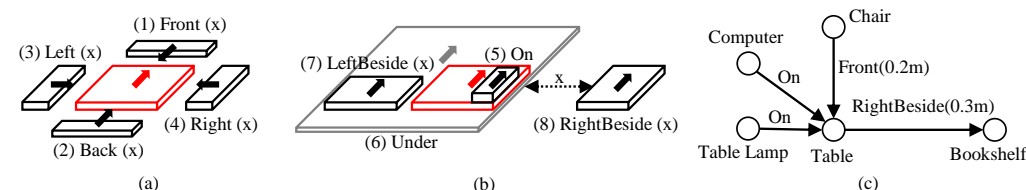

Figure 3: Illustrations of the relations and graphs in Section 3.3. There are eight types of pairwise relations between objects, where a subordinate object either faces toward (a) or shares the same direction as (b) the anchor object (Red). Six of these relations also require an "x" attribute specifying the buffer distance between the two objects. (c): An example of a graph representing multiple relations in a group. The furniture can be organized in topological order as long as the graph is valid.

the sample inputs and outputs to the context to enhance the LLM's results (this approach is also applied to the subsequent two steps).

**Response.** The LLM is instructed to strictly follow a parsable format (i.e., JSON) and ensure the required information is provided in the text fields, detailing each zone's (1) function(s), (2) rectangular size, and (3) furniture list containing names indicating the furniture categories.

**Postprocessing.** We try to find a valid placement for the zones within the room, ensuring no collisions occur. Given that the zones have regular shapes and are relatively few, we explore all placement possibilities using the depth-first search. The zone rectangles are sequentially placed adjacent to corners, walls, or other zones, with backtracking employed in case of collisions. When space allows, placing against corners and walls is prioritized to prevent overcrowding.

**Feedback.** Feedback is provided whenever the LLM's response has an incorrect format, lacks required information, or fails the postprocessing check. For the last case, the LLM is instructed to reassess the room space more carefully to give zones that fit. The LLM can also omit less critical functions or integrate multiple functions into a single zone if space is limited.

### 3.3 FORMING FURNITURE GROUPS AND RELATIONS

In scene synthesis, the spatial relations between/among objects are leveraged to ensure objects are arranged plausibly between/among each other. A scene graph is a classic structure for representing these relationships, and it is particularly suitable for the LLM as it only requires pairwise relations (Çelen et al., 2024; Fu et al., 2024; Lin & Mu, 2024). However, generating a global scene graph that associates many furniture objects significantly increases the risk of logical errors (e.g., circuits), misplacements, and collisions, presenting a challenge for the LLM (Li et al., 2024a). To address this, we instruct the LLM to divide the furniture into groups and construct the graph for each group. An edge in the graph corresponds to a pairwise relation, encompassing the spatial relation (e.g., up/down/front/back/left/right) and buffer distance between objects. Figure 3 (a) and (b) give examples of such relations. The sizes of the furniture are also requested.

Unlike the first step (Section 3.2) with only one room, several zones necessitate their own groupings and relations. Since an LLM agent can independently manage each zone, we dispatch multiple agents to process different zones concurrently (one agent for a zone) to accelerate the generation.

**Context.** The LLM is provided with: (1) the furniture list of the zone from Step 1, (2) the function(s) and size of the zone from Step 1, and (3) furniture and function descriptions. The LLM is instructed to carefully comprehend each furniture object's characteristics and associate them with the zone function. Additionally, furniture sizes are asked to accommodate these relations.

**Response.** The response must include: (1) one or multiple groups (each containing one or multiple furniture objects), (2) pairwise relations (as shown in Figure 3) in each group, and (3) furniture sizes.

**Postprocessing.** We first verify the validity of the grouping, i.e., whether each furniture object belongs to and only belongs to one group. Next, we check the relations within each group, constructing a graph unless logical errors like circuits are present. We then place the furniture according to the

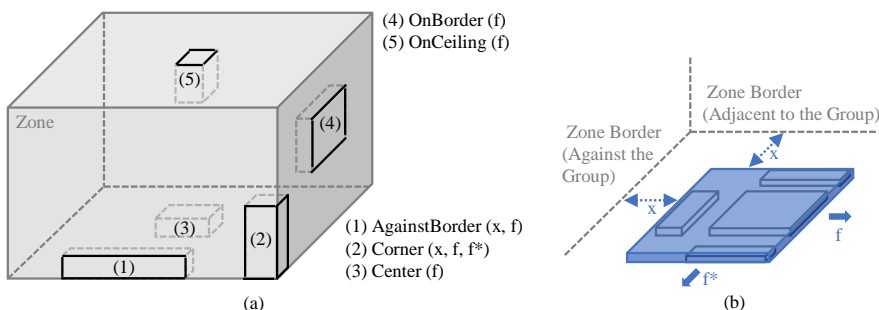

(4) OnBorder (f)
(5) OnCeiling (f)

(1) AgainstBorder (x, f)
(2) Corner (x, f, f*)
(3) Center (f)

Figure 4: Illustrations of anchor rules in Section 3.4. (a): A group can be anchored to a zone through five rules. Each requires an "f" attribute specifying the group's facing direction, while the first two rules also require "x" as the buffer. (b) An example of the "Corner (x, f, f*)" rule. The group (blue bounding box) faces right (i.e., "f" equals "right"), against the opposite-direction border with a buffer distance of "x". "f*" determines the adjacent border that the group aligns with, as a corner is indicated by two perpendicular borders.

relations in each group and check for any collisions. Finally, we check whether all groups can be positioned within the zone without collisions, using an approach similar to that in Section 3.2. If all the preceding checks pass, we select the most appropriate 3D model for each furniture object from a database based on its name (category) and size.

**Feedback.** Errors in groupings or relations are directly addressed in the feedback by specifying which group or relation contains the mistake. If the groups are too crowded for the zone, the LLM is instructed to consider more compact relations (e.g., reduce the buffer distance between objects) or use more miniature furniture while maintaining practicality.

## 3.4 ARRANGING FURNITURE GROUPS

The final task involves arranging the furniture groups generated in Section 3.3 within the zones decided in Section 3.2. Instead of directly assigning positions and orientations to these groups, which often results in out-of-bounds or collided layouts, we instruct the LLM to arrange the groups based on specific anchor rules. These "anchors" include borders, corners, and the ceiling. We also allow the anchor to be the "center" of the zone for more flexible placement. An anchor rule defines a group's relative position/buffer and orientation to the anchor, serving as a spatial relationship between the group and the room, as illustrated in Figure 4. Similar to the above step, we dispatch multiple LLM agents to handle the generation:

**Context.** The LLM is provided with: (1) all furniture groups along with their corresponding furniture names and group sizes, (2) the function(s) and size of the zone, (3) the location and the adjacent walls of the zone, and (4) furniture and function descriptions. The anchor rules are thoroughly explained to the LLM, which is instructed to create a collision-free arrangement that reflects the functions. Even in cases of overcrowded space, unreasonable placements such as a chandelier on the floor or a table mounted on the wall are strictly prohibited.

**Response.** The response must be a list of anchor rules (Figure 4) corresponding to furniture groups.

**Postprocessing.** We verify whether the provided anchor rules can be successfully implemented within the zone. For groups that are not fixed in position (e.g., a group placed in the center), we sample several valid positions and traverse them in a priority-based order: (1) positions that align the group with another group, (2) those align centrally with the anchor, (3) those balance the arrangement within the zone, and (4) those adjacent to other groups.

**Feedback.** If any groups collide, this is reported back to the LLM.

Once the successful arrangements of all zones are complete, the entire scene can be assembled and presented to the user.

(a)                                              (b)

Figure 5: The user interface of our platform. (a) In this example, the user selects four functions and draws an L-shaped room in the left panel before clicking "OK". The platform then generates the corresponding 3D scene and displays it on the right. (b) The user can search for additional furniture and further interact with the generated scene.

## 4 EXPERIMENTS

### 4.1 SYSTEM IMPLEMENTATION

We develop an online 3D platform that integrates our framework. As shown in Figure 5a, the panel to the left enables user interaction, and the canvas to the right displays the 3D scene. After the user selects one or multiple desired functions from the available options, draws the room shape on the point array, and clicks "OK", the platform starts generating the scene that is displayed upon completion. Our platform also supports direct interaction with the 3D scene (Figure 5b). The user can search for suitable furniture, add it to the scene, or remove inappropriate items. Additionally, furniture objects can be adjusted in position, orientation, or scale.

We implement the backend of the framework using Python 3.8 and use GPT-4o, one of the state-of-the-art models of OpenAI, as the LLM. All furniture objects displayed in the scenes are sourced from the Objaverse dataset (Deitke et al., 2023). Our code will be made publicly available.

### 4.2 QUANTITATIVE EVALUATION

In this section, we quantitatively compare our framework with two baseline methods targeting LLM-assisted scene synthesis: LayoutGPT (Feng et al., 2024) and I-Design (Çelen et al., 2024). The evaluation focuses on four aspects: (1) Generation support (whether the method supports **irregular shape** and **user control**). (2) Scene validity, calculating the percentage of **invalid objects** that are either out of bounds or collide with other objects. (3) Text-image alignment measured by the **CLIP score**, indicating how well the generated scenes align with the user inputs. (4) Overall scene quality, assessing the **functionality**, **practicality**, and **aesthetics** judged by **GPT**.

For consistent comparison across methods, we ensure all scenes are generated using GPT-4o, uniformly converted to our platform's format, and rendered under the same configuration. To assess adaptability to different inputs, we generate 500 scenes with varying configurations for our method. Each scene is configured with a random combination of up to 6 functions and a rectangular room shape, with dimensions between 3 and 5 meters. We then generate 500 corresponding scenes using I-Design and LayoutGPT, with the text prompt structured as, "A multi-functional room with the following functions: [function1], [function2], ...". We observe that the official LayoutGPT implementation does not natively support free-formed input, which could lead to an unfair comparison. To address this, we modify its system prompt by inserting the user input above, allowing LayoutGPT to understand the functional needs. Figure 6 showcases scenes generated by all three methods, and Table 1 summarizes the evaluation results.

Our method offers the best generation support among the three methods, allowing for user-defined irregular room shapes and customized functions. Although I-Design enables user refinement through interacting with LLM agents, the room architecture is restricted to a predefined rectangular shape. LayoutGPT, an example-based method, relies mainly on example rectangular rooms from a database and allows control only of the room type and size in its official implementation.

| Functions | Shape | LayoutGPT | I-Design | SceneFunctioner |
|---|---|---|---|---|

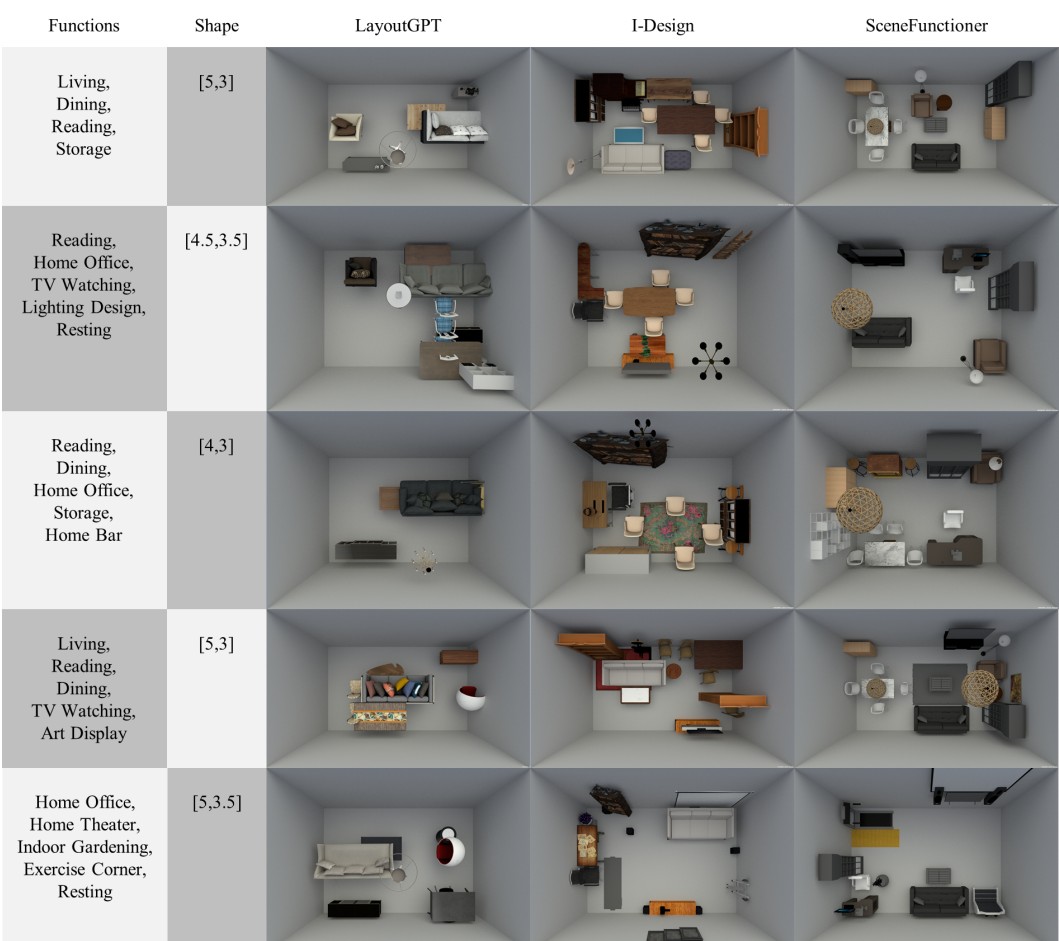

Figure 6: Scenes generated by (a) LayoutGPT, (b) I-Design, and (c) SceneFunctioner. (a) Layout-GPT fails to account for the functions and frequently results in furniture collisions or out-of-bound placements. (b) I-Design successfully accommodates various functions, but the furniture arrangements could cause interference among functions. For example, in the top-row scene, the arrangement is so compact that it blocks pathways for accessing dining, reading, and storage areas. (c) Scene-Functioner effectively balances furniture arrangement with functional needs, achieving the highest overall generation quality among the three methods.

Our method can always generate a valid scene without furniture objects colliding with other objects or with the walls, thanks to the strict verification and feedback mechanism implemented in each step. In contrast, LayoutGPT directly specifies the objects' configuration (i.e., position, orientation, and scale) without checking them, leading to over half of the objects being invalid. I-Design incorporates correction and refinement processes to address such collision cases but cannot eliminate them.

We employ OpenAI's ViT-L/14-336px model (Radford et al., 2021) to compute the CLIP score (cosine similarity multiplied by 100) for the rendered images and their corresponding text descriptions, e.g., "A top-down view of a multi-functional room with the following functions: [function1], [function2], ...". Our method has an advantage over the other two methods, though the scores are relatively close. This result may be due to the limited capacity of the CLIP model, which struggles to effectively encode and correlate functions in the text with visual information in the image. As a result, this metric may not fully capture the performance differences among the methods.

GPT-4o evaluates the last three metrics. For each text prompt, GPT-4o is presented with three images, one from each method, and tasked with selecting the best in terms of three criteria: how well the room accommodates the user's specified functions (**function**), whether the furniture is placed

Table 1: Quantitative evaluation results demonstrate that SceneFunctioner outperforms the two baselines. First, our method supports both irregular room shapes and user control while consistently generating scenes free from furniture collisions or out-of-bound placements. Second, our method achieves the highest CLIP score among all methods, indicating superior alignment between the generated scene (image) and the text prompt. Last, when GPT is tasked with selecting the best scene among the three methods, our method excels in **functionality** and **practicality**, though there is still room for improvement in **aesthetics**.

| Method | Irregular Shape | User Control | Invalid Objects | CLIP Score | GPT-Function | GPT-Practicality | GPT-Aesthetics |
|---|---|---|---|---|---|---|---|
| LayoutGPT | × | limited | 64.71% | 25.83 | 2.0% | 7.6% | 18.6% |
| I-Design | × | ✓ | 1.40% | 27.43 | 46.2% | 24.4% | **41.4**% |
| Ours | ✓ | ✓ | **0**% | **27.60** | **51.8**% | **68.0**% | 40.0% |

Table 2: Participant-rated results for all methods in the first user study. In general, user ratings closely align with GPT ratings. Our method outperforms both baselines in all three criteria.

| Method | User-Function | User-Practicality | User-Aesthetics |
|---|---|---|---|
| LayoutGPT | 0.6% | 9.2% | 13.2% |
| I-Design | 42.0% | 36.8% | 42.2% |
| Ours | **57.4**% | **54.0**% | **44.6**% |

in an accessible and practical layout (**practicality**), and how visually appealing the arrangement is (**aesthetics**). Table 1 lists the percentage of each method rated the best among the 500 sets of scenes. Our method outperforms the baselines in **function** and **practicality**, though slightly underrated in **aesthetics** compared with I-Design. Although I-Design effectively identifies necessary furniture objects, it occasionally fails to arrange them in a way that adequately supports the intended functions. For instance, a bookshelf-chair set intended for reading may be surrounded by furniture for other activities, which can interfere with reading and diminish overall practicality. In contrast, our framework successfully customized the LLM to consistently address functional requirements while ensuring practical arrangements.

## 4.3 USER STUDY

We conduct two user studies to evaluate our framework further. The first study complements the quantitative evaluation by evaluating the same three criteria—**function**, **practicality**, and **aesthetics**—but replaces the GPT evaluator with human participants. We invite twenty-five participants (fifteen males and ten females, with an average age of $\mu = 25.16$ and standard deviation $\sigma = 4.12$). Eleven have experience in art, architectural design, or 3D software. Each participant is randomly assigned twenty sets of images and tasked with selecting the best image in each set. As summarized in Table 2, our method excels across all three criteria, establishing it as a state-of-the-art solution for function-oriented scene synthesis.

The second study involves interacting with our framework on the online platform (see Section 4.1). Each participant is invited to complete three different design tasks, repeating each twice with the same target functions and room shape—once using the generation framework (**assisted**) and once without it (**manual**). The first design has the functions and room shapes specified by our staff, and the participants suggest the latter two. In the manual phase, participants must manually search for, add, and adjust furniture. When assisted by our framework, participants can further refine the generated scene if necessary. In both cases, a scene is considered complete only when the participant and our staff are satisfied with the result. Before the formal experiment, participants receive instructions to ensure they understand the task and are familiar with the platform's operations.

This study uses several metrics to compare the **assisted** and **manual** approaches. First, our platform automatically records **the elapsed time** required to complete scenes. The time includes the entire design process, including interaction and scene generation when using our framework, as well as all manual operations for both approaches. Second, we introduce cross-rating, where two participants

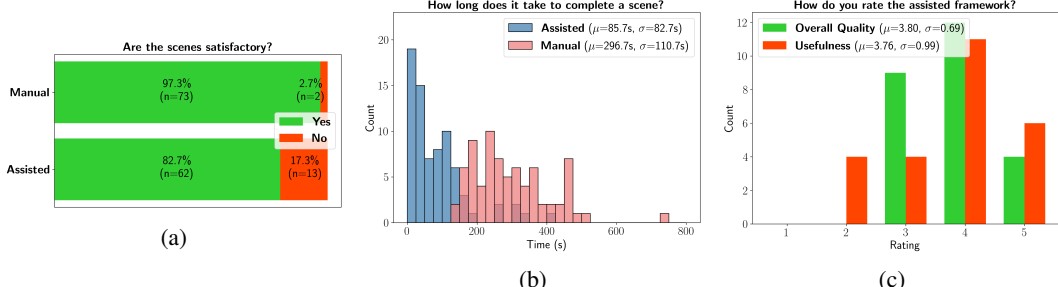

Figure 7: Statistical analysis of the second user study. (a) Although not as preferred as the **manual** scenes, over 80% of scenes by the **assisted** framework are deemed satisfactory by participants. (b) With our **assisted** framework, participants complete each scene in under one and a half minutes on average, manifesting a significant reduction compared with **manual** operations. (c) All participants agree that SceneFunctioner produces scenes of above-average quality. While opinions on its usefulness vary, most participants rate it positively, giving 4 or 5.

work simultaneously, and each judges whether the scene created by the other is **satisfactory** with a simple "Yes" or "No". Lastly, participants rate our framework (**assisted**) based on two criteria: its **usefulness** for scene design and the **overall quality** of the generated scenes, using a 5-point Likert scale.

The same twenty-five users from the first study are invited to participate in this second study, with results illustrated in Figure 7. In the cross-rating, nearly all manual designs are accepted, and most scenes generated by our framework are also considered satisfactory. However, there is a significant difference in the time taken to complete a scene: manual operations averaged nearly five minutes, while scenes assisted by our framework required only 28.9% of that time. Moreover, participants generally rated the quality and usefulness of our framework positively. Several participants with art or architectural backgrounds admire its potential as a valuable tool in indoor design. These results and user feedback indicate that SceneFunctioner significantly enhances design efficiency while delivering above-average quality in function-oriented design.

## 5 Conclusion and Future Work

This paper presents SceneFunctioner, a function-oriented interactive framework leveraging the LLM's capabilities to generate scenes. Following a three-step process, the framework tailors the LLM for zones, furniture groups, and furniture arrangements, given user-specified functions. Quantitative and qualitative experiments demonstrate that our framework consistently generates scenes with appropriate functionality while achieving state-of-the-art quality. However, improvements are still required in the following aspects:

First, our framework does not account for the relations among different zones. Although focusing on generation within individual zones simplifies the task, it occasionally leads to inconsistencies at zone borders. For instance, pathways might be unintentionally blocked. To address this issue, we plan to modify the framework to support consistent generation across zones while still maintaining the task's manageability for the LLM.

Second, the zones are restricted to rectangular shapes, limiting flexibility when dealing with complex room layouts or unconventional furniture arrangements. We previously experimented with irregular shapes but found the LLM's performance significantly reduced, likely due to its current limitations. Nonetheless, we will explore alternative methods for supporting flexible zone shapes.

Finally, there is room for improving both generation quality and efficiency. Currently, the LLM often produces wrong cases, most of which are caught by our framework's postprocessing steps. However, this significantly increases the retries and the overall generation time. Furthermore, certain cases are challenging to detect, e.g., if the LLM places a sofa on top of a table, our framework will follow this placement and produce a poor scene. We are refining the instructions, including more sample inputs and outputs, and enhancing the feedback mechanism to optimize the framework.

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

# A APPENDIX FOR REBUTTAL REVISION

## A.1 SUPPLEMENTARY DETAILS FOR POSTPROCESSING MECHANISMS

In SceneFunctioner, postprocessing phases in both the first and third steps address placing rectangles within a defined space. Specifically, Step 1 involves positioning the zones within the room, while Step 3 focuses on arranging the furniture groups within each zone. Both steps utilize an enhanced version of Algorithm 1, which is a basic template.

---

**Algorithm 1:** The Basic Algorithm for Placing Rectangles within a Space

**Input:** The space $S$ and the rectangles $\Omega = \{R_1, R_2, ..., R_N\}$
**Output:** Whether the rectangles can be placed within $S$, and the placement $X$ (if they can)

1   $W \leftarrow \textbf{All\_Permutations}(\Omega)$;
2   Initialize $X$ as an empty $S$;
3   **for** $\Omega_i = \{R_1^i, R_2^i, ...R_n^i\} \in W$ **do**
4      **if** `Place`$(\Omega_i, 1)$ **then**
5        **return** True, $X$;
6      **end**
7   **end**
8   **return** False, $\emptyset$;
9   **Function** `Place`$(\Omega_i, j)$ **is**
10      $P \leftarrow \textbf{All\_Possible\_Positions}(R_j^i)$;
11      **for** $\mathbf{p} \in P$ **do**
12        **if** $R_j^i$ *can be placed on* $\mathbf{p}$ **then**
13          Place $R_j^i$ in $X$ on the position $\mathbf{p}$;
14          **if** $j = N$ *or* `Place`$(\Omega_i, j+1)$ **then**
15            **return** True;
16          **end**
17          Remove $R_j^i$ from $X$;
18        **end**
19      **end**
20      **return** False;
21   **end**

---

The basic algorithm is exhausts all permutations of the rectangle set (Line 1) and tests each permutation (Lines 3-7). Within each permutation, the iterative subprocess (Lines 9-21) attempts to place each rectangle at all possible positions (Line 10), employing backtracking if placement fails.

To improve its efficiency, we implement several optimization strategies:

1. **Testing a limited number of permutations**: When some rectangles in the input set are identical or nearly identical, permutating them is unnecessary. For cases with many possible permutations, we randomly sample a subset for testing. If all sampled permutations fail, it is unlikely that other permutations will succeed.

Table 3: Performance statistics of SceneFunctioner. Retries do not happen frequently and have limited influence to the overall generation.

| Step | Average Generation Time | Average Time for Retries | Average Retry Count |
|---|---|---|---|
| Step 1 | $7.435s$ | $2.391s$ | 0.628 |
| Step 2 | $9.051s$ | $2.865s$ | 0.449 |
| Step 3 | $6.294s$ | $2.360s$ | 0.876 |

2. **Restricting possible positions**: A large number of potential placement positions could significantly reduce efficiency. In Step 1, only corner positions, including those formed by existing rectangles, are considered. In Step 3, only critical positions are sampled, as described in Section 3.4.

3. **Priority-based attempts**: Step 1 prioritizes placing zones in room corners with the highest number of adjacent walls (Section 3.2). In Step 3, permutations with the order of corner-wall-center will be tested first. Sample positions are also sorted by priority (Section 3.4).

## A.2 FRAMEWORK PERFORMANCE ANALYSIS

We conduct a supplementary performance analysis of each step in our framework, using an additional 500 randomly configured scenes following the setup in Section 4.2 (rectangular rooms with dimensions ranging from 3 to 5 meters and up to 6 functions). The **total generation time** (including retries), **time spent solely on retries**, and **retry count** are recorded, with the average values summarized in Table 3. The statistics demonstrate that, despite the framework's multi-step process and additional mechanisms, the generation efficiency remains generally acceptable to users.

## A.3 GENERALIZABILITY TO MORE SCENES

Our framework demonstrates adaptability to various room scales and shapes in interior design scenarios. Furthermore, with suitable prompts and assets, it can be extended to accommodate additional functions and room categories, such as bedrooms. Illustrative examples of these capabilities are presented in Figure 8.

## A.4 AUGMENTING LAYOUTGPT WITH FUNCTION-BASED EXAMPLES

In Section 4.2, we included functional requirements in the system prompts for LayoutGPT. However, LayoutGPT still selects sample inputs and outputs solely based on room shape, without considering the functional needs. This observation suggests an enhancement to LayoutGPT by incorporating functional considerations when selecting examples.

As suggested by their work, the distance between two rooms with dimensions $[l_1, w_1]$ and $[l_2, w_2]$ is computed using the L2 distance $\|l_1 - l_2\|^2 + \|w_1 - w_2\|^2$. However, calculating a "distance" that accounts for the functions is more complex, as it requires a quantitative representation of a room's functional attributes. We propose an approach for computing such a "function vector" for any scene generated by SceneFunctioner. Let $M$ represent the number of functions, then the function vector for room $L$ can be described as $\mathbf{V_L} = [v_1, v_2, ..., v_M]$, where $v_i \geq 0, \forall i$ and $\sum_i^M v_i = 1$. We use the average value of two components $\mathbf{V_{L,1}}$ and $\mathbf{V_{L,2}}$ to compute $\mathbf{V_L} = \frac{\mathbf{V_{L,1}} + \mathbf{V_{L,2}}}{2}$. The two components are explained as follows:

1. $\mathbf{V_{L,1}}$ **represents the overall function of the furniture within the room.** For a room $L$ with $N_f$ furniture objects $\{f_1, f_2, ..., f_{N_f}\}$, we first retrieve each object's "function vector" $\mathbf{V_f}$ from our annotated data. Each $\mathbf{V_f}$ shares the same representation as $\mathbf{V_L}$. The overall function vector of the room is then computed as $\mathbf{V_{L,1}} = \frac{1}{N_f} \sum_i^{N_f} \mathbf{V_{f_i}}$.

2. $\mathbf{V_{L,2}}$ **reflects the functions in the text prompt used to generate the room.** Let $\{j_1, j_2, ..., j_{N_j}\}$ denotes the indices of the $N_j$ functions described in the prompt, then

| Reading, Home Office, Music Corner, Gaming Zone, Lighting Design | Dining, Reading, Art Display, Indoor Gardening | Reading, Home Office, TV Watching, Art Display | Home Office, Living, Lighting Design, Home Bar |
|---|---|---|---|
| [3.5,2.5] | [3,2] | [3,2] | [3.5,2] |

(a)

| Reading, TV Watching, Living, Dining, Social Gathering, Fireplace | TV Watching, Dining, Reading, Storage, Home Bar, Exercise Corner, Gaming Zone | Dining, Home Office, Living, TV Watching, Resting, Exercise Corner, Storage | Dining, Home Office, Reading, Home Theater, Music Corner, Gaming Zone, Indoor Gardening |
|---|---|---|---|
| [7,5] | [8,6] | [8,6] (Irregular) | [10,9] (Irregular) |

(b)

| Sleeping | Sleeping, TV Watching, Art Display, Indoor Planting | Sleeping, Home Office | Sleeping, Resting, Storage |
|---|---|---|---|
| [4,3] | [5,4] | [4,4] | [4.5,3] |

(c)

Figure 8: Results that showcase SceneFunctioner's generalizability. (a) The framework effectively accommodates multiple functions, even within compact rooms. (b) For large-scale, complex rooms, the proposed zoning approach enables management of numerous functions. (c) SceneFunctioner is also capable of generating functional and visually appealing bedroom designs.

$\mathbf{V_{L,2}} = \frac{1}{N_j} \sum_i^{N_j} \mathbf{I_{j_i}}$, where $\mathbf{I_x}$ is a unit vector with only the $x$-th element set to 1. This component evenly distributes the functions across the vector.

With the proposed metric, we can account for both the room shape and the function vector when selecting examples. Given a room $L$ with dimensions $[l, w]$ and function vector $\mathbf{V_L}$, and a user query requesting a room with dimensions $[l', w']$ and functions $\mathbf{V_{L'}} = \mathbf{V_{L',2}}$, the total distance $d$ is computed as Equation 1 (weight $\alpha = 0.5$):

$$d = \alpha\sqrt{\|l - l'\|^2 + \|w - w'\|^2} + \|\mathbf{V_L} - \mathbf{V_{L'}}\|_1 \qquad (1)$$

Table 4: Comparing the augmented LayoutGPT with SceneFunctioner. We observe a significant improvement in LayoutGPT's performance. However, it still struggles with addressing furniture collisions, and scene quality remains outperformed by ours. This reinforces the strong performance of our framework in delivering both functional and practical scene designs.

| Method | Invalid Objects | CLIP Score | GPT-Function | GPT-Practicality | GPT-Aesthetics |
|---|---|---|---|---|---|
| LayoutGPT-Augmented | 33.46% | 26.71 | 17.5% | 25.5% | 34.0% |
| Ours | **0%** | **27.53** | **82.5%** | **74.5%** | **66.0%** |

We configure LayoutGPT with an example dataset consisting of 1000 scenes generated by our framework and compare it with our framework. 200 new input prompts, using the configuration in Section 4.2, are randomly created for instructing a new batch of scenes with both methods. For each scene, LayoutGPT is provided with the ten most similar scenes from the example dataset, based on the minimum distance $d$. Additionally, we ensure that the system prompts are updated accordingly. Figure 9 showcases scenes generated by LayoutGPT, alongside the corresponding example scenes.

The results for the quantitative evaluation, similar to that in Section 4.2, are summarized in Table 4. By selecting appropriate examples from our dataset, the performance of LayoutGPT is significantly improved. However, there are still many instances where it fails to handle object collisions. Additionally, SceneFunctioner continues to outperform LayoutGPT in function, practicality, and aesthetics scores. While providing relevant examples (sample inputs/outputs) can enhance the LLM's performance, we suggest that current LLMs still require substantial improvement to directly deduce layouts involving multiple objects and complex restrictions. This highlights the ongoing need for task-specific tailoring.

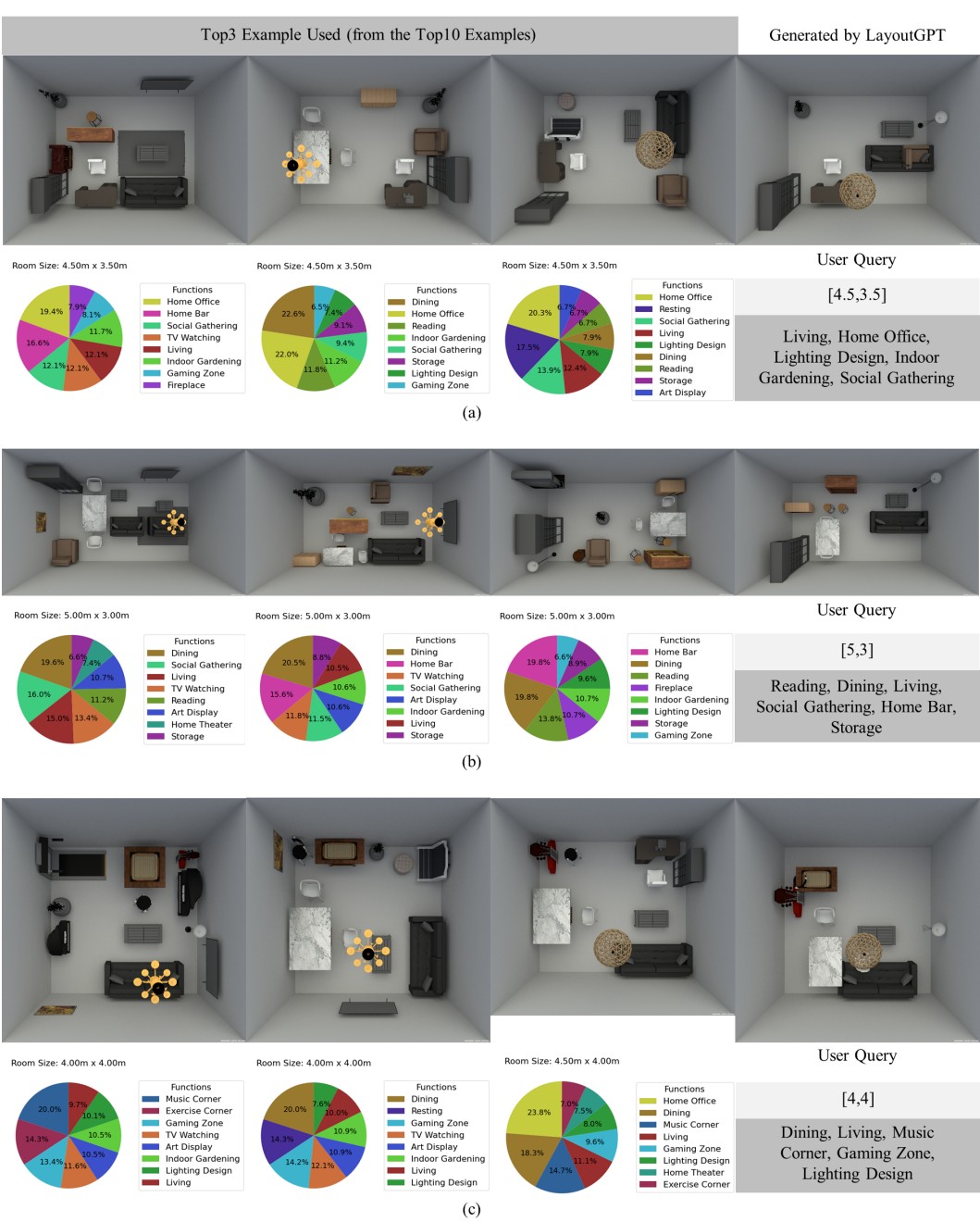

Figure 9: Overview of the augmented LayoutGPT. For each user query, the augmented method selects the ten most similar scenes from our dataset. In each row, the left three images shows the top three examples along with their features, including a visualized function vector (pie chart) and room shape. The rightmost image presents the generated result by LayoutGPT. While the inclusion of function-based examples noticeably enhances generation quality, issues such as frequent collisions and irrational layouts persist.

