# OpenReview forum: "SceneFunctioner: Tailoring Large Language Model for Function-Oriented Interactive Scene Synthesis"
_ICLR.cc/2025/Conference — Submitted to ICLR 2025_

### Official Review · Reviewer_2Daj · 2024-11-01

**Soundness:** 3
**Presentation:** 3
**Contribution:** 3
**Rating:** 8
**Confidence:** 3

**Summary:**

The paper introduces a novel approach for the interactive synthesis of 3D scenes, called SceneFunctioner. The framework allows users to sketch rooms and to define functional zones. Given the user input, an LLM is used to generate furniture layouts for each zone based on a three-step generation process (zone creation, local furniture grouping, and final arrangement). A feedback mechanism is used to manage potential LLM errors that may occur during the generation process (e.g. incorrect formatting, object collision, etc,). The paper discusses quantitative results and provides user studies that show that SceneFunctioner outperforms previous approaches.

**Strengths:**

- The three-step generation process (zoning, local grouping, and final arrangement) and the introduced feedback mechanisms for the LLMs are novel and interesting.
- The approach focuses on room functions, which is not addressed by other  approaches.
- The conducted user studies seem reasonable and validate the effectiveness of SceneFunctioner.
- The paper is well-written and easy to follow.

**Weaknesses:**

- I appreciate the conducted user study, however the validation of the approach appears to be more on the lightweight side.
- While the use of zones seems to generate interesting results, it is unclear if the zones can be easily adapted for more complex concepts and layouts (e.g. mixed-use spaces).
- The current approach does not consider the aesthetics of the generated furniture layouts (which has also been highlighted as limitation).

**Questions:**

- How does SceneFunctioner handle function conflicts in multi-functional zones?
- How would an LLM perform better for irregular room layouts (e.g. how would it need to be trained to achieve better performance)?
- How would this framework adapt to advancements in LLM capabilities (e.g. what features would be required to improve upon the current results)?
- How adaptable is SceneFunctioner to user feedback after an initial scene is generated (if a user wants minor adjustments, would the framework require a complete re-synthesis)?
- Is there any limitations regarding the number of different zones? E.g. could SceneFunctioner easily be extended to also cover outdoor spaces?
- The authors may want to also consider the following work as part of their related work discussion: R. Ma, A. Gadi Patil, M. Fisher, M. Li, S. Pirk, B.-S. Hua, S.-K. Yeung, X. Tong, L. Guibas, H. Zhang, Language-Driven Synthesis of 3D Scenes from Scene Databases, ACM Transactions on Graphics (Proceedings of SIGGRAPH Asia), 2018

**Details Of Ethics Concerns:**

No concerns.

---

> ### Author Response · Authors · 2024-11-15
>
> We thank you for your valuable comments on our work. Here we provide our point-by-point reply.
>
> We look forward to your response and appreciate it if you could share your ideas.
>
> **1.I appreciate the conducted user study, however the validation of the approach appears to be more on the lightweight side.**
>
> We understand the concern about the comparison experiments, but we believe they are appropriately done. Evaluation of scene quality (functionality, aesthetics, etc.) is inherently subjective and lacks reliable quantitative metrics. Without reference images and other data, computing the collision rates of objects and the CLIP score was what we could do. We also supplemented three criteria (function, practicality, and aesthetics), evaluating them with both a GPT evaluator and human participants.
>
> **2.While the use of zones seems to generate interesting results, it is unclear if the zones can be easily adapted for more complex concepts and layouts (e.g. mixed-use spaces).**
>
> The motivation for proposing ‘zones’ is handling more complex concepts/layouts and multi-use spaces, since it helps to decompose the concepts and functions, lightens the burden for the LLM, and enhances understanding of the task. It is still possible that ‘zones’ could fail in extremely large and complex spaces, but we believe they have the generalizability to most interior scenarios in real life. If you have more insights or questions, please discuss them with us.
>
> **3.The current approach does not consider the aesthetics of the generated furniture layouts (which has also been highlighted as limitation).**
>
> The current framework has some aesthetical limitations as noted in the paper. However, we have instructed the LLM to consider aesthetics (e.g., balance among objects and coherence in layout) through prompting in each step, and our experiments have demonstrated that the results are aesthetically appealing.
>
> **4.How does SceneFunctioner handle function conflicts in multi-functional zones?**
>
> We appreciate that you have raised an interesting point. Currently, we would say that no mechanism is implemented to explicitly solve functional conflicts, but we expect the first step of our framework to never provide such cases. A ‘zone’, as described in our paper, should ensure consistency of the functions within it. The LLM is explicitly informed of this and carefully instructed to coordinate the functions. Still, we acknowledge that understanding and handling relationships among functions are worth studying, rendering it for our future work.
>
> **5.How would an LLM perform better for irregular room layouts (e.g. how would it need to be trained to achieve better performance)?**
>
> In our paper, the LLM is instructed to decide the zone sizes while the framework automatically places them within the room. For better performance in irregular spaces, the LLM could be provided with a ‘reference’ rectangular division plan. For example, an L-shaped room could be divided into two or three rectangles. Giving the references as prompts can reduce the possibilities of zone collision errors, but may likely lead to zone sizes that are similar to the division plan (which is undesirable). Therefore, currently we only provide these prompts when encountering continuous failures.
>
> Another solution is to dispatch an LLM agent that focuses only on room division when determining the zones. However, overlooking whether the functions can fit in these zones may eventually lead to worse results.
>
> **6.How would this framework adapt to advancements in LLM capabilities (e.g. what features would be required to improve upon the current results)?**
>
> This is another interesting perspective. We would expect major advancements and adjustments in our framework when provided with more capable LLMs: (1) Ideally, the LLM could directly suggest the zone positions and shapes (could be irregular) in the first step so no postprocessing is required. The functions will also be allocated more appropriately. (2) The second and the third steps can possibly be combined if the LLM can handle all furniture relations and layouts in a single shot. More complicated relations and layouts may also be possible. (3) More factors (e.g., lighting, windows and doors, and human interactions) may be more easily included into the framework as long as they are appropriately prompted for the LLM to understand them.
>
> **7.How adaptable is SceneFunctioner to user feedback after an initial scene is generated (if a user wants minor adjustments, would the framework require a complete re-synthesis)?**
>
> Currently, our framework would need a complete re-synthesis for adjustments. For minor adjustments, the user may want to directly interact with the generated scene using our 3D scene platform. Of course, we appreciate the idea of enabling adjustment features and will investigate a stepwise and context-based ‘editable’ framework in the future.
>
> **(due to the character limit, we will add another comment)**

---

> > ### Author Response · Authors · 2024-11-15
> >
> > **8.Is there any limitations regarding the number of different zones? E.g. could SceneFunctioner easily be extended to also cover outdoor spaces?**
> >
> > Considering the computational complexity of postprocessing in step 1, we currently limit the number of zones to 10. For SceneFunctioner as it currently is, it struggles to handle a large number of zones. However, we believe that with modifications in configurations and approaches, our coarse-to-fine framework can be extended to address object layouts in outdoor spaces. For example, when placing the vegetation or artificial elements, we can consider determining ‘zones’ based on design objectives, followed by grouping and arranging them. We have to clarify that our solution may be inspiring for outdoor scene generation, but such problems could not be easily ‘covered’ due to a bunch of factors (e.g., terrain, sunlight, temperature).
> >
> > **9.The authors may want to also consider the following work as part of their related work discussion: R. Ma, A. Gadi Patil, M. Fisher, M. Li, S. Pirk, B.-S. Hua, S.-K. Yeung, X. Tong, L. Guibas, H. Zhang, Language-Driven Synthesis of 3D Scenes from Scene Databases, ACM Transactions on Graphics (Proceedings of SIGGRAPH Asia), 2018**
> >
> > Thank you for pointing out this missing reference. After going through this paper, we think it is worth discussing as a related work (which will be addressed in the revision file).

---

> ### Author Response · Authors · 2024-11-17
> **Revised paper uploaded**
>
> We have just uploaded a rebuttal revised version of our paper, with all changes marked in blue for your convenience. This revision file supplements the related work in Section 2.1, as you suggested in #9. We would greatly appreciate it if you could review the updated manuscript and let us know if it addresses some of your concerns.

---

> > ### Comment · Reviewer_2Daj · 2024-11-25
> > **Reply**
> >
> > Dear authors, thank you for your answers to my questions and for preparing the revised version of the manuscript. You answers helped me to further understand your method and to clarify my concerns. I am now more convinced about this work and I am willing to increase my score.

---

> > > ### Author Response · Authors · 2024-11-26
> > > **Thanks for your feedback**
> > >
> > > Thank you very much for your thoughtful feedback and for raising your score. Your comments are invaluable in helping us improve the paper.

---

### Official Review · Reviewer_6nMz · 2024-11-02

**Soundness:** 3
**Presentation:** 3
**Contribution:** 3
**Rating:** 8
**Confidence:** 4

**Summary:**

This paper introduces SceneFunctioner, a function-oriented scene synthesis framework that allows users to decide on room functions and room shape. The three-stage framework employs LLMs at each stage, which first divides the scene into zones containing the furniture for one or more functions. The second stage further divides the furniture within each zone into groups and constructs a graph for relations within each group. The last step places the groups within each zone to complete the layout. Quantitative results and user studies prove the effectiveness of the proposed approach.

**Strengths:**

a) The overall idea of coarse-to-fine scene synthesis strategy is interesting and novel. b) Scene synthesis based on room functions yields plausible and realistic object arrangements. c) The interactive scene synthesis framework allows users to design customized spaces with ease. d) The visual results look good.

**Weaknesses:**

a) In L231-232, the author mentions appending sample inputs and outputs to the context. How are these samples selected/formed? LayoutGPT selects the closest samples for in-context learning using room dimensions & type for generating random layouts of specific room types. However, synthesizing scenes based on room functions is a more specific task than generating random plausible layouts. Therefore, for a fair comparison, I suggest providing LayoutGPT with in-context samples that are closest in terms of room functions. b) The author mentions that the wrong results produced by LLMs increase the retries and the overall generation time. I am curious about the quantitative analysis of the feedback mechanism. How much time do the retries take when compared to the actual generation time? How many retries are required on average per-stage? c) While the paper presents intriguing qualitative results, it would be interesting to see how the method performs on larger rooms (dimensions more than 5 meters) or smaller rooms, different room types (such as bedrooms) to better evaluate the method's generalizibility.

**Questions:**

All of my questions are listed in the weaknesses section, and I may adjust the rating if they are well addressed.

---

> ### Author Response · Authors · 2024-11-15
>
> We thank you for your valuable comments on our work. Here we provide our point-by-point reply to your questions. We will post the additional results in the revision file when ready.
>
> We look forward to your response and appreciate it if you could share your ideas.
>
> **1.In L231-232, the author mentions appending sample inputs and outputs to the context. How are these samples selected/formed? LayoutGPT selects the closest samples for in-context learning using room dimensions & type for generating random layouts of specific room types. However, synthesizing scenes based on room functions is a more specific task than generating random plausible layouts. Therefore, for a fair comparison, I suggest providing LayoutGPT with in-context samples that are closest in terms of room functions.
> Sample inputs/outputs to our work are manually designed.**
>
> Regarding the suggestion on LayoutGPT, we appreciate your valuable insight and agree that function-oriented examples could be fairer. However, the challenge lies in defining the ‘distance’ of functions. As far as we know, embedding and representation of functions in interior 3D scenes is still an unsettled problem (and this paper is investigating it). The results may vary significantly with different standards for deciding the examples. Additionally, sample inputs also need to be modified to suggest the functions. We are actively working on this and may post the results to the revision file. We also look forward to your response if you have any ideas on it.
>
> **2.The author mentions that the wrong results produced by LLMs increase the retries and the overall generation time. I am curious about the quantitative analysis of the feedback mechanism. How much time do the retries take when compared to the actual generation time? How many retries are required on average per-stage?**
>
> Thank you for your interest. We will soon list these retry-related statistics in the revision file.
>
> **3.While the paper presents intriguing qualitative results, it would be interesting to see how the method performs on larger rooms (dimensions more than 5 meters) or smaller rooms, different room types (such as bedrooms) to better evaluate the method's generalizibility.**
>
> Thank you for your interest. Our framework can be generalized to such configurations, and we will soon post the results to the revision file.

---

> ### Author Response · Authors · 2024-11-17
> **Revised paper uploaded**
>
> We have just uploaded a rebuttal revised version of our paper, with all changes marked in blue for your convenience. Specifically, Section A.2 provides statistics for all steps in our framework, including generation time and retry counts. Section A.3 showcases our framework's generalizability to smaller/larger rooms and a different room type (bedroom). We are actively working on addressing LayoutGPT's function-based examples, as you suggested. We would greatly appreciate it if you could review the updated manuscript and let us know if it addresses some of your concerns.

---

> > ### Comment · Reviewer_6nMz · 2024-11-25
> >
> > Thank you for responding to my questions and your patience. I appreciate the responses regarding the generation time and the generalizability to different room types.
> >
> > Comparison with LayoutGPT (can be performed on a small set of scenes): Considering you have "annotated data suggesting appropriate furniture for these functions", a distance/similarity metric can be defined to select in-context samples based on the ratio of furniture corresponding to these target functions or something similar. While this metric may not be a perfect solution, it would provide a more intuitive distribution to the baseline method.

---

> > > ### Author Response · Authors · 2024-11-26
> > > **Thank you for your feedback**
> > >
> > > Thank you for your thoughtful feedback. We fully agree with your suggestions and are already working on comparative experiments using a similar approach. Specifically, we use scenes generated by SceneFunctioner as the example dataset for LayoutGPT. Leveraging the annotated data of furniture, we can efficiently compute a function vector for each scene, representing its functional attributes (e.g., "60% Living + 25% Dining + 15% Office").
> > >
> > >
> > > To determine the distance metric when selecting examples, we compute a weighted sum of two components: (1) The L1 distance of the function vectors. (2) The room shape distance (used in LayoutGPT).
> > >
> > >
> > > We believe this approach will provide insightful comparisons. A revised paper, supplementing these results, will be uploaded soon.

---

> > > ### Author Response · Authors · 2024-11-27
> > > **Revised paper (version 2) uploaded**
> > >
> > > We have just uploaded the second rebuttal revised version of our paper, with all changes marked in **purple** for your convenience. We would greatly appreciate it if you could review **Section A.4** of the updated manuscript and let us know if it addresses your first question. If you have any additional questions or comments regarding our work, we would be glad to hear from you.

---

> > > > ### Comment · Reviewer_6nMz · 2024-12-01
> > > >
> > > > Thank you for performing this comparison. With this, my concerns have been resolved. I have increased my score.

---

> > > > > ### Author Response · Authors · 2024-12-02
> > > > > **Thanks for your feedback**
> > > > >
> > > > > Thank you very much for your thoughtful feedback and for raising your score. Your comments are invaluable in helping us improve the paper.

---

### Official Review · Reviewer_jrDz · 2024-11-03

**Soundness:** 2
**Presentation:** 3
**Contribution:** 1
**Rating:** 3
**Confidence:** 4

**Summary:**

Given a list of user-specified scene functions and a room shape, the paper proposes an
interactive framework based on LLMs to synthesize a scene that adheres to the given scene
functionalities.

The main gaps identified in the literature that motivates this work are: (a) scene generation methods that are functionality-unaware, focusing only on object relations, and (b) LLMs requiring contextual information to generate functionally-aware scenes.

This work presents an interactive, human-in-the-loop scene generation method, with user-specified functionalities and room shapes. The proposed method can be divided into three stages: (1) dividing a scene into “zones” based on user selected functionalities, (2) organize furniture for each “zone” into groups, and (3) placing furniture at the right orientation at an appropriate location. In every stage, there exists a verification and feedback mechanism to address potential issues arising out of LLM errors. This feedback mechanism from the humans makes the approach more reliable despite using LLMs. In terms of quantitative evaluations, four different measures are recorded. They are: (1) Generation support: a binary indicator that tells whether the framework allows support for irregular shape and user control, (2) Percentage of invalid objects that are out of bound or collide with other objects, (3) CLIP-score that measures how well the generated scene aligns with given input, and (4) Overall scene quality measured in terms of functionality, practicality and aesthetics
judged by GPT. For comparison, the paper looks at LayoutGPT (NeurIPS 2023) and I-Design (arXiv 2024), both of which do not incorporate human feedback.

To summarize again, the input is a set of user-specified scene functionalities, scene shape. The output is a scene populated with furniture. The dataset used in ObjaVerse. The LLM chosen here is GPT-4o.  There is no learning mechanism involved (i.e., no training  step involved) as the paper proposes a better way to prompt LLM by dividing the task of scene synthesis into aforementioned three steps, instead of a single one.

**Strengths:**

* Well-written paper

* Dividing the scene synthesis task into three stages to make things relatively easy for LLM is an interesting way to stabilize LLMs for complex tasks in 3D

* The second user study shows that the proposed framework is beneficial to the designers.

**Weaknesses:**

* The only contribution is a stepwise human-in-the-loop prompting technique to use an LLM (GPT-4o)
for scene synthesis. Such a framework does not necessarily address the issue of generating functionally plausible and usable 3D scenes. While there are some merits to the study in the paper in terms of helping LLMs adapt to 3D scene generation tasks,  the overall impact and utility is limited.

* In addition, the experiments presented in the paper are performed on a small dataset (500 scenes) that is manually generated for this framework.

Overall, the paper proposes a step-wise LLM prompting strategy for scene synthesis. The idea is
interesting but lacks technical contribution.

**Questions:**

* Did the users have a pool of scene functions to choose from? If not, the presented approach is going to see high input-entropy. How does the LLM handle that? If yes, how many total number of scene functions are provided to the users to choose from?  I do see in Line 366 which mentions that each scene is configured with randomly selected 6 functions. From how many available functions are these chosen from?

* Were there any quantitative metrics used to verify/validate the correctness of scenes generated at each *stage*? For example, zone allocation accuracy for stage 1, furniture grouping coherence for stage 2, and placement accuracy for stage 3. Such a quantitative evaluation for every stage can potentially provide more insights into the framework on where things are better/worse.

* Finetuned LLMs have shown to provide good generalization capability. Current framework uses the GPT-4o LLM as is. It would be interesting to see the results when the LLM is finetuned for scene synthesis task and then used to synthesize a new scene. I was actually hoping to see some sort of finetuning, foloowed by an contextual analysis. That would have been more interesting.

* The paper uses the words “scene” and “room” interchangeably. Is the method proposed in paper for individual room synthesis or scene-level (entire house) synthesis? Please adhere to one convention: either use scene or room everywhere. Better yet, good to provide a clarifying statement in the Intro and/or methodology section about this.

---

> ### Author Response · Authors · 2024-11-15
>
> We thank you for your valuable comments on our work. Here we provide our point-by-point reply. We understand your concerns, but we respectfully disagree with you on some points, **especially concerning your misunderstanding of our framework in #1 and #2.**
>
> We look forward to your response.
>
> **1.The only contribution is a stepwise human-in-the-loop prompting technique to use an LLM (GPT-4o) for scene synthesis. Such a framework does not necessarily address the issue of generating functionally plausible and usable 3D scenes. While there are some merits to the study in the paper in terms of helping LLMs adapt to 3D scene generation tasks, the overall impact and utility is limited.**
>
> We have to emphasize that our framework is not a human-in-the-loop technique. The user acts before the generation starts, and can interact with the scene or restart after the generation finishes. The postprocessing and feedback mechanisms are automatically performed. We will soon provide more technical details on these steps in the revision file, and we think that these mechanisms also contribute to the method. Also, we think the proposed ‘zone’, as the bridge for abstract representations (function) and spatial representation (layout), is also inspiring for the research field of scene representation.
>
> Regarding the utility, the experiments demonstrated our framework generally outperforms LayoutGPT and I-Design and achieves decent quality. The user study suggested that our framework saves much time for interior design. Overall, we believe that our framework appropriately addresses generating functionally plausible and usable scenes.
>
> **2.In addition, the experiments presented in the paper are performed on a small dataset (500 scenes) that is manually generated for this framework.**
>
> The ‘manually generated’ is another misunderstanding, since all scenes for all three methods are automatically generated, with random-sampled function-specified prompts. We also believe that a sample number of 500 is sufficient to show our framework’s relative performance.
>
> **3.Did the users have a pool of scene functions to choose from? If not, the presented approach is going to see high input-entropy. How does the LLM handle that? If yes, how many total number of scene functions are provided to the users to choose from? I do see in Line 366 which mentions that each scene is configured with randomly selected 6 functions. From how many available functions are these chosen from?**
>
> Yes, there is a pool containing 16 functions to choose from.
>
> **4.Were there any quantitative metrics used to verify/validate the correctness of scenes generated at each stage? For example, zone allocation accuracy for stage 1, furniture grouping coherence for stage 2, and placement accuracy for stage 3. Such a quantitative evaluation for every stage can potentially provide more insights into the framework on where things are better/worse.**
>
> Thanks for your valuable insight. Although we cannot quantify whether a step output is ‘proper’ or ‘good’, we can judge whether it ‘succeeds’ or results in ‘violation’, as it is what the postprocessing mechanism does. We will soon provide a revision file supplementing these statistics.
>
> **5.Finetuned LLMs have shown to provide good generalization capability. Current framework uses the GPT-4o LLM as is. It would be interesting to see the results when the LLM is finetuned for scene synthesis task and then used to synthesize a new scene. I was actually hoping to see some sort of finetuning, followed by an contextual analysis. That would have been more interesting.**
>
> Thank you for your suggestion. Due to our limited time and budget, we found it difficult to collect (construct and label) sufficient data for effective finetuning but instead focused on improving the prompting and sample inputs/outputs. Nevertheless, we agree that finetuning is a promising approach in our problem setting that may further enhance the generation quality and efficiency.
>
> **6.The paper uses the words “scene” and “room” interchangeably. Is the method proposed in paper for individual room synthesis or scene-level (entire house) synthesis? Please adhere to one convention: either use scene or room everywhere. Better yet, good to provide a clarifying statement in the Intro and/or methodology section about this.**
>
> Our paper targets individual room synthesis and both ‘scene’ and ‘room’ refer to this. We will clarify such terms and avoid ambiguity.

---

> ### Author Response · Authors · 2024-11-17
> **Revised paper uploaded**
>
> We have just uploaded a rebuttal revised version of our paper, with all changes marked in blue for your convenience. Specifically, Section A.1 includes supplementary technical details for postprocessing steps. Section A.2 provides statistics for all steps in our framework. Additionally, we add a sentence in the Introduction Section to clarify 'scene' and 'room'. We would greatly appreciate it if you could review the updated manuscript and let us know if it addresses some of your concerns.

---

> > ### Comment · Reviewer_jrDz · 2024-11-23
> >
> > Authors, thank you for responding to my questions and for your patience in waiting to hear back from me.
> >
> > I've read through the rebuttal and below are my responses, numbered based on the rebuttal.
> >
> > 1-A: If the post-processing and feedback are automatically done, can you elaborate on how do you check for "lack of required information (line 242-243)", how is LLM instructed to reassess (line 244), 2) how do you determine validity of graphs and correctness of relationships (line 269, 294), 3) how do you measure whether the furniture arrangement is "too crowded" (line 295), 4) how do you determine the scale of the furniture (how miniature the furniture should be)? (line 296). The text needs to be made more clear on how these tasks are preformed automatically (without human interference). With that, I would like to take back the first strength of the paper "well-written paper". The main paper lacks many important details which are necessary for readers to understand the proposed system. If these details are part of supplementary material, it should be incorporated into the main paper.
> >
> > #1-B: The concept of abstract and spatial scene representation has been in use in research community working on 3D layout generation. Agreed that the use of term "zone" for abstract representation is new.
> >
> > #2-A: The proposed system is about function-oriented scene synthesis and the word "manually" meant for choosing the 16 functionalities, are manually selected (& which are not benchmarked) and used for scene generation.
> >
> > #2-B: The scene dataset used for experiments is explicitly generated from scratch for this paper, which may prove to be a useful scene dataset if released (instead of using existing 3D indoor scene datasets, say 3D-FRONT or Scene-instruction Pair Dataset proposed in InstructScene, Lin and Mu 2024). I am not clear on why Objaverse dataset for furniture objects is used. That is atypical.
> >
> > #4: How do you judge whether a step succeeds or is in violation? Again, is this done automatically or by a human looking at tghe outputs of those intermediate steps? That leads to my next question -- if you cannot quantify, then how do you use "success"/"in-violation" judgement to provide feedback, "automatically". If you have written a code that checks "success"/"in-violation", it can be used as a binary output and reported as how many of 500 scenes succeed in step-1/2/3.
> >
> > Overall, the rebuttal does not satisfactorily address my concerns (rather, raises more). Help clarify my mis-understanding(s), if any. Looking forward to the discussion.

---

> > > ### Author Response · Authors · 2024-11-24
> > > **Thank you for your Response**
> > >
> > > We sincerely appreciate your review and thoughtful feedback on our rebuttal. We are pleased to see your active engagement and questions regarding our work. Below, we provide a response to address your concerns and clarify the raised issues. We look forward to further discussions if you have additional concerns or suggestions.
> > >
> > > ## #1-A
> > > **Our scripts automatically check for “lack of required information”**: As explained in lines 234-235, the “required information” is what we ask the LLM to respond. Since the LLM responds in a JSON format, any missing field can be easily checked.
> > >
> > > **We instruct the LLM to reassess**: ```The provided zones <zone_information> cannot fit in the room with <room_architecture>. For reference, the room can be divided into <a_division_plan>. Please carefully consider zone sizes that can fit in this room.```
> > >
> > > **Our scripts automatically check the validity of graphs and relations**: In the directed graph connecting furniture nodes (like Figure 3(c)), if any circuit (regardless of the edge direction) exists, then the graph is invalid. For example, a structure like "A->B, A->C, B->C" is invalid. Additionally, if the local arrangement leads to object collision, it is considered invalid.
> > >
> > > **Our scripts automatically check for “too crowded”**: A furniture group, as a whole, might not be able to fit in a zone if the local arrangement takes up too much space. For example, a group taking up 2.2m\*1.5m cannot fit in a 2m\*2m zone.
> > >
> > > **We instruct the LLM to determine the scale of furniture**: ```The furniture group <group_information> takes up <space_occupation> that cannot fit in the zone <zone_information>. Please design more compact relations to arrange the furniture, or use smaller furniture to reduce the space occupation.```
> > >
> > > **Regarding the details**: Certain aspects may require further clarification or elaboration, as raised by you. However, given that all reviewers have rated the presentation of our paper as “good,” we believe these points can be addressed with a few revisions. We would be delighted to hear any additional questions you may have and will actively revise our paper.
> > >
> > > ## #1-B
> > > Thank you for your acknowledgment.
> > >
> > > ## #2-A
> > > The functions were manually chosen to provide a demonstration of SceneFunctioner’s capabilities. However, we have shown that our framework is not constrained to these specific functions and can adapt excellently to other scenarios, such as bedrooms (refer to Section A.3). Our paper addresses a novel topic that focuses on functional tailoring. Therefore, we argue that the manual selection of functions should not be a point of concern.
> > >
> > > ## #2-B
> > > We agree that existing datasets like 3D-FRONT are suitable for typical scene synthesis research. However, the unbalanced and limited object categories in these datasets render them unsuitable for our study, which emphasizes functional diversity. For instance, 3D-FRONT provides numerous instances of common objects like beds, tables, and chairs but lacks items like musical instruments, game consoles, and plant stands.
> > >
> > > On the other hand, Objaverse offers diverse furniture instances covering nearly all conceivable functions. Additionally, we would not consider the use of Objaverse "atypical," as several recent studies [1-4] have also employed it.
> > >
> > > [1] Fu, R., Wen, Z., Liu, Z., & Sridhar, S. (2025). Anyhome: Open-vocabulary generation of structured and textured 3d homes.
> > >
> > > [2] Yang, Y., Sun, F. Y., Weihs, L., VanderBilt, E., Herrasti, A., Han, W., ... & Clark, C. (2024). Holodeck: Language guided generation of 3d embodied ai environments.
> > >
> > > [3] Çelen, A., Han, G., Schindler, K., Van Gool, L., Armeni, I., Obukhov, A., & Wang, X. (2024). I-design: Personalized llm interior designer.
> > >
> > > [4] Aguina-Kang, R., Gumin, M., Han, D. H., Morris, S., Yoo, S. J., Ganeshan, A., ... & Ritchie, D. (2024). Open-Universe Indoor Scene Generation using LLM Program Synthesis and Uncurated Object Databases.
> > >
> > > ## #4
> > > **We have to emphasize again that these postprocessing checks are fully automated and require no human intervention.** Basically, all “violations” are collisions, so the steps are "successful" only under: In Step 1, zones fit inside the room without collisions. In Step 2, furniture is arranged locally within groups without collisions, and each group fits within its designated zone. In Step 3, furniture groups fit within the zones without collisions.
> > >
> > > Regarding success rates, we conducted experiments on 500 scenes, without retrying after any failure caused by "violations." Among these scenes: 81 failed in Step 1, 57 failed in Step 2, and 144 failed in Step 3. This resulted in only 218 successful cases overall. Assuming independence between steps, the individual success rates are 83.8%, 86.4%, and 60.2% for the three steps, respectively.
> > >
> > > For more details, please refer to Section A.2 of the revised paper, which examines generation time and average retry counts.

---

> > > > ### Comment · Reviewer_jrDz · 2024-12-02
> > > >
> > > > Authors, I have posted a comment in Reviewer *ryvl*'s thread. There exist grave concerns with the technical exposition of the paper and unfortunately, to me, the paper is not a suitable fit for ICLR. I remain negative with my score.

---

### Official Review · Reviewer_ryvL · 2024-11-04

**Soundness:** 2
**Presentation:** 3
**Contribution:** 1
**Rating:** 1
**Confidence:** 5

**Summary:**

The paper introduces SceneFunctioner, an interactive framework that leverages GPT-4o for function-oriented 3D indoor scene synthesis, addressing the gap in user-specific, practical room designs. Unlike existing methods, it focuses on distributing user-selected functions into distinct zones and organizing furniture to align with these functional requirements while preventing common issues like object collisions and logical inconsistencies. The framework follows a structured three-step process—deciding zones, forming furniture groups, and arranging them within zones—supplemented by robust postprocessing and feedback mechanisms. The main contributions include a structured approach to function-oriented scene synthesis, the introduction of zones to manage complexity, and iterative verification steps to ensure collision-free, practical designs that align with user inputs.

**Strengths:**

The strengths of the paper include its approach to integrating function-oriented design in 3D scene synthesis, ensuring that generated layouts align with user-specified functional needs. It introduces a multi-step process involving zone-based division and iterative postprocessing checks to prevent furniture collisions and logical inconsistencies. The framework’s use of GPT-4o demonstrates strong LLM capabilities, effectively balancing user input with practical design outcomes. Additionally, its feedback loop ensures error correction and refinement, resulting in high-quality, customized scene synthesis with enhanced user interaction and reduced design time.

**Weaknesses:**

This paper reads more like a technical report of a user application and prompt engineering rather than addressing fundamental research problems.

The proposed solution also seems tentative, as demonstrated by several limitations:
- Limited zone shapes: Restricted to rectangular zones and room shapes, reducing flexibility for complex layouts.
- Zone border inconsistencies: Does not account for relationships between adjacent zones, potentially causing pathway issues.
- Generation inefficiencies: Requires multiple retries due to LLM errors, increasing overall generation time.
- Not considering floor plan structures: Overlooks the placement of doors and windows, which can significantly affect the usability of the generated scenes.

The paper tackles a niche problem that may not resonate with a broad ICLR audience.

The use of GPT, a closed-source model, means the solutions are less reliable and more tentative due to the lack of understanding and access to the model’s workings.

The spatial understanding of LLMs has been extensively studied in prior research, and this paper does not provide new insights that could inspire or motivate future work in this area.

**Questions:**

NA

---

> ### Author Response · Authors · 2024-11-15
>
> We thank you for your valuable comments on our work. Here we provide our point-by-point reply. We understand your concerns, but we respectfully disagree with you on some points.
>
> **Your reviews #3 and #5 contradict each other, where ‘a niche problem’ seems controversial to ‘extensively studied’. We are confused about the reviews.**
>
> We look forward to your response.
>
> **1.This paper reads more like a technical report of a user application and prompt engineering rather than addressing fundamental research problems.**
>
> Our paper addresses a research problem of ‘tailoring user preferences to scene representation and generation’. It may not be ‘fundamental’ as you suggest, but we consider it a valuable topic for deepening the understanding of representation for 3D scenes. Also, our contributions are not limited to prompt engineering. We believe our coarse-to-fine framework and the proposed ‘zone’ representation provide new insights into 3D scene representation and related research.
>
> **2.The proposed solution also seems tentative, as demonstrated by several limitations**
>
> We acknowledge that our solution requires improvements. However, some limitations mentioned are not critical or can be addressed by expanding our framework:
>
> Limited zone shapes: Flexible shapes can be supported since we can simply ask the LLM to support them. However, as noted in the paper, the LLM’s performance may significantly reduce. Thus, we find it more reliable to support flexible zone layouts rather than the shapes themselves.
>
> Generation inefficiencies: As long as there are LLM errors, extra efforts are needed. For example, I-Design[1] relies on a ‘layout corrector’ to fix such errors. We have implemented postprocessing mechanisms to handle them and minimize their impact on the generation time. Also, the average time for generating each scene is less than 30 seconds, which is acceptable for live user interaction. For comparison, I-Design requires about two minutes to generate each scene.
>
> Floor plan structures: These factors can be easily addressed by expanding each step to include them.
>
> [1] Çelen, A., Han, G., Schindler, K., Van Gool, L., Armeni, I., Obukhov, A., & Wang, X. (2024). I-design: Personalized llm interior designer. arXiv preprint arXiv:2404.02838.
>
> **3.The paper tackles a niche problem that may not resonate with a broad ICLR audience.**
>
> Our research falls into general topics like ‘applications of 3D scenes’ and ‘scene representation and understanding’, which are appropriate topics for ICLR. The research problem is worth studying (as you also suggest in #5 that extensive prior research exists in this area), regardless of whether it is ‘broad’.
>
> **4.The use of GPT, a closed-source model, means the solutions are less reliable and more tentative due to the lack of understanding and access to the model’s workings.**
>
> First, our research does not emphasize the mechanisms of LLMs. Instead, we emphasize how LLM understands and yields 3D scenes. Second, we do not think that a close-source model means the method is less reliable. This statement lacks factual evidence and theoretical basis. According to our experiments and existing research, the result scenes are reliable.
>
> Please refer to our experimental sections.
>
> **5.The spatial understanding of LLMs has been extensively studied in prior research, and this paper does not provide new insights that could inspire or motivate future work in this area.**
>
> How we address the functions of the scene distinguish our work from prior research. We have proposed using ‘zones’ as a bridge for abstract representations (function) and spatial representation (layout), which is novel. Additionally, we believe the coarse-to-fine framework provides instructive inspiration in addressing more scene representation and generation problems.
>
> Please refer to our ‘related works’ section and our supplementary video.

---

> > ### Author Response · Authors · 2024-11-17
> > **Revised paper uploaded**
> >
> > We have just uploaded a rebuttal revised version of our paper, with all changes marked in blue for your convenience. We would greatly appreciate it if you could review the updated manuscript and let us know if it addresses some of your concerns.

---

> ### Comment · Reviewer_ryvL · 2024-11-25
>
> Thank you for your reply.
>
> “Functionality preference” is a niche problem primarily relevant to very specific user applications. The general spatial understanding of LLMs (e.g., whether LLMs have a notion of relative positioning, collisions, etc)—problems shared by a wide range of downstream applications (not limited to digital design, robotics, SLAM, and more)—has been extensively studied. However, “functionality preference” does not contribute significantly to the broader spatial understanding of LLMs that is relevant to larger ICLR communities. It resembles more of a product pain point rather than a research problem.
>
> I disagree with the assertion that the mentioned limitations are not critical. From a product pain point perspective, issues such as limited zone shapes, zone border inconsistencies, generation inefficiencies, and floor plan structures can negatively impact the user experience just as much as functionality preference. It is not fair to claim that only the addressed problems are critical.
>
> These problems are not as readily addressable as suggested by the authors. As noted by the authors, accommodating flexible shapes could reduce the LLM’s performance. Additionally, the current pipeline requires multiple retries to avoid collisions. If the proposed method enables LLMs to generate valid layouts (e.g., no collisions, no blocked pathways) in a single attempt without relying on post-processing, I would champion the paper without hesitation. Clearly, this is an important and non-trivial problem shared by a much larger community. If the authors genuinely believe these issues can be addressed, it is strongly recommended to resolve them for resubmission.
>
> Regarding the use of GPT-4o, the issue is not its usage but rather the sole reliance on it for validation. What happens if OpenAI updates their model checkpoint? Would the claims and experiments presented in the paper remain relevant? It is strongly recommended that the authors demonstrate their approach on multiple LLMs, preferably using open-source models.
>
> Unfortunately, all of my concerns remain unaddressed.

---

> > ### Author Response · Authors · 2024-11-28
> >
> > Thank you for your reply. Unfortunately, we have to maintain a different perspective from you on certain points.
> >
> > ### #1
> > You focus on arguing that our work is more of an “application” or “product” problem rather than a research problem. While we acknowledge the value of industrial applications and their relevance, we maintain that your claim is appropriate. We have clearly framed our work as a research problem and highlighted its contributions to the field. All other reviewers have recognized the novelty of our approach in the research context.
> >
> > Additionally, you mention that our paper is not “broad” enough to appeal to a wide audience. While we respect differing opinions on what constitutes “general” applicability, we believe it is important to note that not all valuable research needs to address a broad problem. For example, there are numerous works nowadays for fine-tuning and tailing LLMs for very specific tasks. On https://paperswithcode.com/, you can see over 5000 different tasks. The opinion that our work may not address a general problem does not diminish our contribution. As already stated, our proposal of ‘zones’ and the coarse-to-fine framework may inspire future work across a wider range of related tasks.
> >
> > ### #2
> > It seems that you are raising a perspective of “product pain” again, which, in the context of a research paper, is inherently inappropriate. The focus should not be “user experience” as we are not presenting a product. Even from a user experience standpoint, it is unclear whether the limitations you mention would genuinely impact the user experience. In fact, as evidenced by the user study presented in our paper, most participants were satisfied with the results produced by our framework.
> >
> > ### #3
> > We acknowledge that certain limitations, as listed in our paper, remain unaddressed at this stage. However, we believe these limitations do not substantially affect the performance or the overall contribution of SceneFunctioner. We have also outlined potential solutions for some issues in future work.
> >
> > We agree that such a single-attempt, error-free generation framework would represent a breakthrough. However, this is not a limitation of our framework. Our framework does not aim for a “perfect” system, and the inclusion of postprocessing and feedback mechanisms stems from the understanding that current LLMs are prone to errors. Therefore, it is not appropriate to critique our framework solely based on its inability to achieve error-free, no-retry generation.
> >
> > ### #4
> > We appreciate your suggestion to evaluate our approach using different LLMs. However, we believe that using GPT-4o is sufficient to effectively demonstrate SceneFunctioner’s contributions, workflow, and results.

---

> > ### Comment · Reviewer_jrDz · 2024-12-02
> >
> > Thank you, Reviewer *ryvl*. This is reviewer jrDz. I wanted to reply to this thread to second your concerns and issues raised with this paper.
> >
> > Even after looking at the revised upload of the paper, I do not see a major technical contribution from the paper. The whole paper seems to be centered around "Functionality preference" and how one can engineer an LLM to help navigate this aspect of 3D scenes for scene synthesis. What new insight does the proposed system give to the community? Look at the paper I-Design (https://atcelen.github.io/I-Design/) -- this paper tackles a similar problem, but falls short of giving a "Technical Contribution", which is also prevalent in I-Design. I-Design actually take free-form language input, whereas this paper does not even support that. What I see this paper do is to take template prompts and pre-defined "functional zones" as input (as shown in Figure 6). Most importantly, there is no learning involved (including I-Design).
> >
> > If merely trying to engineer, which the paper calls "tailoring", LLMs is the main theme of the paper, not to mention multiple retires to get to a plausible 3D scene, it is, at best, a technical report.
> >
> > As Reviewer *ryvl* mentioned, what would be impressive is that the LLM enables plausible and useable scene synthesis in a single try. That would mean you are endowing the language model with spatial understanding of the world around it, which means, there is an attempt at addressing the problem on the technical front. Simply making different functional zones and having a set of functions to select from, and involving the user for such selections (making it "interactive") is ornamental to the paper, and anyone reading the paper will fail miserably to find any meaningful technical contribution.
> >
> > At the ICLR level, this paper is extremely unsuitable, and I am going to keep my score to Reject (3).

---

> ### Comment · Reviewer_ryvL · 2024-12-02
>
> I agree that not every paper needs to address a general problem, and not every solution must be free of limitations. However, these factors collectively influence the contribution of the paper.
>
> I maintain my position that the technical contribution of this paper is highly incremental, given the narrow scope of the targeted problem, the limited solution, and, equally importantly, the insufficient validation, which leads to **irreproducibility**.
>
> If OpenAI updates their model checkpoint, since no one has access to the original model, all the claims and experiments presented in the paper will become irrelevant. This seriously affects the reproducibility of the method. Evaluation solely on a closed-source model is far from sufficient.

---

### Author Response · Authors · 2024-11-22
**Follow-Up on Revision and Response**

Dear Reviewers,

We hope this message finds you well. A few days have passed since we submitted our reply and revision file. We wanted to check if our responses and updates adequately address your concerns.

If you have any additional questions or comments regarding our work, we would be glad to hear from you. We look forward to your feedback and appreciate your time and consideration.

Best regards

---

### Author Response · Authors · 2024-12-01
**Invitation for Further Discussion and Feedback**

Dear Reviewers,

With the extended discussion period, we still have a few days to address any additional questions or comments you may have regarding our work. We would greatly appreciate any further feedback and are eager to engage in further discussion. Thank you for your time and thoughtful consideration.

Best regards

---

### Meta-Review · Area_Chair_3HG7 · 2024-12-17

**Metareview:**

In this paper, the authors introduce SceneFunctioner, an interactive scene synthesis framework that leverages LLM to prioritize functional requirements of indoor scene synthesis. The framework is user-friendly, allowing users to select specific functions and room shapes. It employs a novel three-step coarse-to-fine generation process for scene synthesis. The generated results are visually appealing and functionally consistent with the user inputs. Overall, the paper is well-written and presents an interesting approach.

However, there are concerns from all reviewers regarding the technical contributions and limitations of the work. Firstly, the framework primarily utilizes LLMs without significant modifications or specialized training to address the problem of scene synthesis. This reliance on existing models raises questions about the novelty and innovation of the approach. Additionally, the functionalities are not mathematically well-defined and appear limited in scope, which may hinder reproducibility and the ability for others to build upon this work. Furthermore, the method is constrained to using only 2D rectangular shapes for furniture and zones during synthesis, limiting its applicability to more complex or realistic scenarios.

While the paper presents an intriguing application of LLMs in interactive scene synthesis, it would benefit from deeper technical contributions and a more robust methodology. Therefore, I don't recommend the paper.

**Additional Comments On Reviewer Discussion:**

During the rebuttal phase, the authors addressed most of the questions posed by the reviewers. However, there are several key issues raised by Reviewers jrDz and ryvL that were not adequately addressed by the authors. I agree with the concerns they expressed, and the other reviewers share this viewpoint as well in the end, which has led to my final decision.

---

### Decision · Program_Chairs · 2025-01-22

Reject